# Use of a Silkworm (*Bombyx mori*) Larvae By-Product for the Treatment of Atopic Dermatitis: Inhibition of NF-κB Nuclear Translocation and MAPK Signaling

**DOI:** 10.3390/nu15071775

**Published:** 2023-04-05

**Authors:** Meiqi Fan, Young-Jin Choi, Nishala Erandi Wedamulla, Qun Zhang, Seong Wan Kim, Sung Moon Bae, Young-Seek Seok, Eun-Kyung Kim

**Affiliations:** 1Division of Food Bioscience, College of Biomedical and Health Sciences, Konkuk University, Chungju 27478, Republic of Korea; fanmeiqi@kku.ac.kr; 2Department of Food Science and Nutrition, College of Health Science, Dong-A University, Busan 49315, Republic of Korea; choiyoung11@donga.ac.kr (Y.-J.C.); 2178445@donga.ac.kr (N.E.W.); 2273245@donga.ac.kr (Q.Z.); 3Center for Silver-targeted Biomaterials, Brain Busan 21 Plus Program, Dong-A University, Busan 49315, Republic of Korea; 4Department of Health Sciences, The Graduate School of Dong-A University, Busan 49315, Republic of Korea; 5Department of Food Science and Technology, Faculty of Animal Science and Export Agriculture, Uva Wellassa University, Badulla 90000, Sri Lanka; 6Department of Agricultural Biology, National Institute of Agricultural Sciences, Rural Development Administration, Wanju Gun 24226, Republic of Korea; tarupa@korea.kr; 7Gyeongnam Agricultural Research and Extension Services, Jinju 52733, Republic of Korea; smbae@korea.kr; 8The Province of Gangwon Agricultural Product Registered Seed Station, Chuncheon 24226, Republic of Korea; air5738@korea.kr; 9Center for Food & Bio Innovation, Dong-A University, Busan 49315, Republic of Korea

**Keywords:** atopic dermatitis, *Bombyx mori*, Yeonnokjam, silkworm by-product, NF-κB p65

## Abstract

Atopic dermatitis (AD) is a long-lasting inflammatory skin disease that contributes to the global health burden and impacts 10–20% of the world’s population. In this study, we determined the anti-AD effect of a by-product of silkworm (*Bombyx mori*) larval powder, strain Yeonnokjam (SLPY), as a sustainable, natural source for the development of therapeutic agents for AD. HaCaT cells were used to assess the in vitro anti-inflammatory activity of SLPY, and a 1-chloro-2,4-dinitrobenzene (DNCB)-induced mouse model was used to study the in vivo anti-AD effects. SLPY treatment downregulated the expression of the inflammatory cytokines TNF-α, IL1β, IL-8, and Cox-2 in stimulated HaCaT cells. Similarly, the topical application of SLPY in DNCB-treated mice downregulated the expression of inflammatory cytokines and proteins while ameliorating the clinical features of AD. Further, SLPY treatment inhibited the nuclear translocation of NF-κb p65, thereby supporting the efficacy of SLPY in the treatment of AD.

## 1. Introduction

Over the past few decades, atopic dermatitis (AD), also known as atopic eczema, has gained attention owing to its increasing prevalence worldwide. Although this disease is commonly diagnosed in children, adults are also potential targets. Atopic dermatitis affects approximately 6–20% of children around the world [1]. The pathophysiology of this multifactorial disease remains incompletely explored [2]. Atopic dermatitis is commonly characterized by persistent pruritus, lichenification, skin pain, scaling and peeling of the skin, and dyspigmentation [1,2]. Protein deficiency, climate, air pollution, food allergies, and obesity are major risk factors associated with AD [3]. In contrast, a skewed balance of T-helper (Th) cells and skin barrier dysfunction are the main contributing factors to AD-like skin lesion development. In addition to abnormal immune responses, immunoglobulin (Ig) E-mediated allergic reactions also play a major role in AD [4]. Accordingly, mast cell IgE receptor-linked allergen activation plays a central role in the pathogenesis of AD [4]. Moreover, increased quantities of eosinophils, basophils, and dendritic cells are characteristic features of inflamed skin in AD [3].

The skin serves as a wall between the internal and external environments, providing protection against foreign agents. Stratum corneum and tight junctions are barrier structures that prevent external penetration. Transmembrane proteins, such as claudin and occludin, and tight junction plaque proteins, such as zonula occludens (ZO), are the major components of tight junctions [5]. These tight junction proteins have also been associated with AD. Decreased claudin-1 protein expression is strongly associated with AD severity. Abnormal filaggrin expression is also prominent in patients with AD [6]. In addition to skin barrier dysfunction, the key role of Th2 cells in AD has been well documented in relation to the immune response. Effector Th lymphocytes, Th 2, are formed by the naïve lymphocytes and produce an array of cytokines including IL-4, IL-5, IL-6, IL-9, IL-10, and IL-13 [7]. Moreover, Th1 cells play an important role in the chronic stages of AD. Thus, the expression of interferon-gamma (IFN-γ) significantly increases [8].

The rapid growth of the world’s population has led to an attempt to increase food production, mainly by targeting protein-based foods. Consequently, the exploration of novel and sustainable food sources is urgently required. Recently, edible insects have been identified as potential alternative protein resources owing to their diverse functional properties [9]. Among the insect protein sources, the functional properties of silkworm (*Bombyx mori*) powder are well known [10,11]. Several studies have investigated the various protocols for producing silkworm powder with improved ingestion and functional properties [10]. Steamed and freeze-dried silkworm powder has gained considerable attention owing to its promising amino acid profile and essential minerals and omega-3 fatty acid content. Thus, it has been suggested as a potential source of food supplements [11]. However, the maturity stage and variety significantly influence the nutrient composition and functional properties of silkworm powder [10,11,12,13]. Silkworm powder has several pharmacological effects. The alleviating effect of silkworm powder on alcoholic fatty liver disease has been well studied [14,15]. Ji et al. [14] showed that silkworm powder effectively alleviates hepatic steatosis and injury in ethanol-treated rats through several mechanisms, including the inhibition of oxidative stress and inflammatory response. Moreover, studies have reported the inhibitory effects of steamed and freeze-dried silkworm powder (SFSP) on hepatic fibrosis and hepatocellular carcinoma [16]. The anti-melanogenic activity of SFSP [17] and the ameliorative effect of mature silkworm powder on scopolamine-induced amnesia [18] have been well documented.

Although many studies have indicated the functional properties of SFSP, studies exploring the anti-AD effects of steamed and freeze-dried by-products of mature silkworm larval powder are still in their infancy. In the present study, we used the silkworm strain Yeonnokjam, a newly bred high-yield strain with promising functional properties [19]. To our knowledge, this is the first study to utilize the silkworm by-product of the Yeonnokjam strain to assess its potential ameliorating effect on AD. We investigated the anti-AD effects of steamed and freeze-dried mature silkworm larval powder of the Yeonnokjam strain using a DNCB-induced mouse model. The anti-AD effect of mature silkworm larval powder was assessed by analyzing the expression profiles of inflammation-related cytokines and histopathological characteristics. The results of the study revealed the significantly low expression levels of tumor necrosis factor (TNF)-α, IL-4, IL-6, IL-10, IL-13, and IL-17 in DNCB-induced mice treated with the powder by-product of mature silkworm larvae, suggesting its potential in the management of AD.

## 2. Materials and Methods

### 2.1. Materials

The steamed and freeze-dried powdered by-product of mature silkworm from the Yeonnokjam strain (SLPY), Golden Silk (SLPG), and Baekokjam (SLPB) was kindly provided by the rural development administration (Wanju Gun, Republic of Korea). Olive oil was procured from a supermarket (South Korea). DNCB was purchased from Sigma-Aldrich (St. Louis, MO, USA). Antibodies against β-actin (sc-47778), TNF-α (sc-52746), inducible nitric oxide synthase (iNOS; sc-7271), and cyclooxygenase-2 (COX2; sc-376861) were purchased from Santa Cruz Biotechnology (Santa Cruz, CA, USA). Antibodies against p-p38 (9215S), p-38 (9212S), p-ERK (4377S), and ERK (4695S) were purchased from Cell Signaling Technology (Danvers, MA, USA). Horseradish peroxidase (HRP)-conjugated goat anti-rabbit IgG, HRP-conjugated goat anti-mouse IgG, and FSD™ 594 conjugated goat anti-rabbit IgG and and FSD™ 488 conjugated goat anti-mouse IgG were acquired from BioActs (Incheon, Korea).

### 2.2. Preparation of Powdered by-Products of Mature Silkworm Larvae

Silkworm larvae at the 5th instar 8-day stage were soaked in distilled water at room temperature (25 ± 2 °C) for 60 min. After washing using distilled water, the larvae were soaked in boiling water for 5 s and steamed for 120 min. The extract discharged from the silkworm larvae during the steaming process was collected and frozen at −60 °C. The frozen extract was freeze-dried to obtain the powdered silkworm larval by-product.

### 2.3. Treatment of HaCaT Cells with Powdered Silkworm Larval by-Products

HaCaT cells were obtained from the Korean Cell Bank (Seoul, Republic of Korea) and cultured in Dulbecco’s modified Eagle’s medium supplemented with penicillin–streptomycin (1%; GIBCO, Grand Island, NY, USA) and fetal bovine serum (10%; Hyclone, Logan, UT, USA). Cultured cells were kept at 37 °C under humidified conditions with a 5% CO_2_ supply. HaCaT cells were seeded in 1 × 10^5^ cells/mL cell density. Following incubation for 24 h, the cells were treated with the powdered by-product of mature silkworm larvae: SLPY, SLPG, and SLPB, at 100 and 200 µg/mL concentration levels. After 30 min of sample treatment, the cells were stimulated with TNF-α (10 ng/mL) and IFN-γ (10 ng/mL) for 6 h.

Considering the efficacy of SLPY, the stimulated HaCaT cells were tested with SLPY (50, 100, and 200 µg/mL) for 6 h to determine the inflammation-related protein expression levels.

### 2.4. Reverse Transcription–Quantitative Polymerase Chain Reaction

RNA extraction was carried out using TRIzol reagent (Sigma-Aldrich, St. Louis, MO, USA) in accordance with the instructions provided by the manufacturer. Following RNA extraction, a DeNovix spectrophotometer (PhileKorea, Republic of Korea) was used to measure the concentration and purity of the RNA. A Luna Universal qPCR kit for cDNA synthesis (BioLab, MA, USA) was used to the synthesize first-strand complementary DNA (cDNA). Reverse transcription–quantitative polymerase chain reaction (RT-qPCR) was performed in triplicate according to a previously reported study [20], using *GAPDH* as an internal reference gene. The relative mRNA expression was calculated using the equation:
Relative mRNA expression = 2^−(∆Ct of target gene − ∆Ct of *GAPDH*)^


Table 1 lists the primer sequences employed in this study.

### 2.5. Cell Viability

HaCaT cells were cultured at 5 × 10^3^ cells/mL cell density in a 96-well plate, and the cell viability was determined using 3-(4,5-dimethylthiazol-2-yl)-2,5-diphenyltetrazolium bromide (MTT; Promega, Madison, WI, USA). After incubating for 24 h, the HaCaT cells were treated with 50, 100, 200, or 400 µg/mL of SLPY, followed by the addition of MTT solution, and incubated at 37 °C for 4 h. Next, the supernatant was decanted, and dimethyl sulfoxide was added to dissolve the formed formazan crystals. Absorbance was recorded at 540 nm.

### 2.6. Immunofluorescence Analysis

To determine the NF-κB nuclear translocation, the HaCaT cells were cultured (5 × 10^3^) in an 8-well slide chamber (SPL Life Science Co., Seoul, Republic of Korea). After a 24h incubation period, the cells were stimulated with TNF-α and IFN-γ at 10 ng/mL, followed by treatment with powdered by-products of silkworm larvae at concentrations of 100 and 200 μg/mL for 6 h. The HaCaT cells were then fixed with ice-cold methanol and permeabilized in 0.1% Triton X-100 for 15 min. After blocking with 5% normal goat serum, the cells were incubated with antibodies against NF-κB (1:300 dilution) and α-tubulin (1:300 dilution) at 4 °C. The cells were then incubated with the corresponding goat anti-rabbit IgG (FSD 594, 1: 1000 dilution) and goat anti-mouse IgG (FSD 488, 1: 1000 dilution) for 1 h. The slides were examined using a Zeiss 700 confocal microscope (Zeiss, Oberkochen, Germany) at ×400 magnification, after adding a mounting medium containing DAPI.

### 2.7. Western Blotting

HaCaT cells and dorsal skin tissues were blended with lysis buffer supplemented with protease inhibitors (Roche, Mannheim, Germany) and centrifuged for 20 min at 4 °C at 15,928× *g*. The supernatant was separated and the protein concentration was determined. Western blot analysis was performed according to the method described in Lee et al. [21]. Protein expression levels of TNF-α, Cox2 and iNOS were assessed using Western blotting with dorsal skin tissue.

### 2.8. Stimulation of AD-like Skin Lesion and an Experimental Animal Model

Seven-week-old female BALB/c mice (18 ± 2 g) were purchased from Nara Biotech Co., Ltd. (Pyeongtaek, Republic of Korea) and housed in a pathogen-free environment at 20–21 °C with 40–45% relative humidity and a 12 h light/dark cycle. BALB/c mice were placed separately in transparent plastic cages bedded with aspen chips and provided free access to tap water and a standard mouse diet. All of the experiments conformed to the requirements of the Dong-A University Animal Care and Use Committee (DIACUC-2138).

AD-like skin injuries were stimulated in 8-week-old female BALB/c mice treated with 1% DNCB. The mice were randomly allocated to six groups (*n* = 6 per group) after a one-week acclimation period. The treatment groups were as follows: (1) control group without any treatment (CON), (2) DNCB-induced group (AD), (3) DNCB treated + Vehicle (ADV), (4) DNCB treated + Vehicle + 50 mg/kg SLPY (SLPY-L), (5) DNCB treated + Vehicle + 100 mg/kg SLPY (SLPY-H), (6) DNCB treated + Vehicle + Dermatop 0.25% (prednicarbate 2.5mg/g; Handok, Gangnam-gu, Republic of Korea) (D). The dorsal hair of the mice was shaved (approximately 2 × 3 cm) using an electric shaver before DNCB application. Then, the sensitization was performed with 200 μL of 1% DNCB, prepared by dissolving DNCB in an acetone–olive oil mixture (3:1 *v/v*). Four days after DNCB sensitization, the same volume (200 μL) of 0.5% DNCB was applied to challenge the dorsal skin in two-day intervals for three weeks. SLPY homogenized in the vehicle was topically applied to dorsal skin three times a week for three weeks. Following three weeks of treatment, the mice were sacrificed and blood samples were collected. The collected blood samples were centrifuged at 4 °C for 20 min at 848.2× *g* to separate the serum, which was stored at −80 °C until further use. The skin tissue was isolated for RNA extraction, Western blot, and histopathological analysis.

### 2.9. RT-qPCR of RNA Extracted from DNCB-Induced AD Mice

RNA was isolated from the mouse dorsal skin tissues using TRIzol reagent (Sigma-Aldrich, St. Louis, MO, USA), complying with the instructions provided by the manufacturer. RT-qPCR analysis was conducted as described in Section 2.4. Appendix A shows the primer sequences for the genes employed in the current study.

### 2.10. Histological Observation of DNCB-Treated AD Mice

Dorsal skin dissected from the SLPY-treated AD mice was fixed with 10% formaldehyde for 24 h. Fixed tissues were embedded in paraffin prior to sectioning into 4 µm thick slices. Then, hematoxylin and eosin (H&E) was used to stain the sectioned tissues for morphological observation. Mast cell infiltration was determined by toluidine blue staining. An optical microscope (Leica, Wetzlar, Germany) was employed to capture the stained tissue sections at ×50 magnification.

### 2.11. Serum Immunoglobulin Analysis of DNCB-Induced AD Mice

Mouse IgG2a and IgE enzyme-linked immunosorbent assay kits from Bethyl Laboratories Inc. (Montgomery, TX, USA) were used to measure the serum IgG2a and IgE levels in the mice treated with SLPY, respectively. All experiments were performed following the instructions specified by the manufacturer.

### 2.12. Amino Acid Analysis

The amino acid separation of the silkworm larval powder was carried out using an L-8900 amino acid analyzer (Hitachi High-Technologies, Tokyo, Japan) connected to a 4.6 mm ID × 60 mm column packed with a Hitachi custom ion exchange resin. The injection volume was 20 µL.

### 2.13. Statistical Analysis

SPSS version 11.5 for Windows (SPSS Inc., Chicago, IL, USA) was used to analyze the data. The mean values of at least three replicates were compared using one-way analysis of variance using Duncan’s post hoc tests. Values of *p* < 0.05, *p* < 0.01, *p* < 0.001, and *p* < 0.0001 were considered statistically significant.

## 3. Results

### 3.1. Powdered by-Products of Silkworm Larvae Exhibit Anti-Inflammatory Effects on TNF-α/IFN-γ-Treated HaCaT Cells

The HaCaT cells were co-stimulated with the pro-inflammatory mediators TNF-α (10 ng/mL) and IFN-γ (10 ng/mL) to assess the anti-inflammatory effect of powdered silkworm larvae by-products. Three silkworm by-products, namely, Yeonnokjam, Golden Silk, and Baekokjam, were treated at 100 and 200 µg/mL for 6 h. Figure 1 illustrates the effect of silkworm by-products on the expression levels of inflammatory cytokines in TNF-α/IFN-γ-co-treated HaCaT cells. All of the silkworm by-products (SLPG, SLPY, and SLPB) significantly (*p* < 0.01) downregulated the TNF-α expression levels. There was no significant difference between the expression levels of TNF-α between the two concentration levels. Additionally, the expression level of IFN-γ also followed the same pattern and was significantly downregulated by all of the silkworm by-products. Further, a low concentration (100 µg/mL) of SLPY significantly downregulated the expression level of IL-8 compared to that achieved with SLPG and SLPB at 100 µg/mL. However, the SLPG and SLPB-treated cells exhibited significantly (*p* < 0.05) lower IL-8 expression levels. Similarly, low concentrations of SLPY significantly (*p* < 0.01) downregulated the expression of Cox-2 compared to that achieved by SLPG and SLPB. However, the Cox-2 expression levels of the SLPY-treated cells (100 and 200 µg/mL) were insignificantly different (Figure 1D). Therefore, even at low concentrations, SLPY markedly decreased the expression level of the inflammatory cytokines. Owing to the promising effect of SLPY in attenuating inflammatory cytokine expression, SLPY was selected for further in vivo studies in DNCB-treated mice.

### 3.2. Percentage of Cell Viability after SLPY Treatment

The HaCaT cells were tested against SLPY at 50, 100, 200, and 400 µg/mL concentrations to assess the cell toxicity of SLPY. The results are shown in Figure 1E. There was no cell toxicity effect exhibited up to a concentration of 400 µg/mL. Therefore, SLPY exhibited no obvious toxicity in the HaCaT cells.

### 3.3. Powdered by-Products of Silkworm Larvae Attenuate NF-kB p65 Nuclear Translocation in Stimulated HaCaT Cells

The immunofluorescence results of TNF-α/INF-γ co-stimulated HaCaT cells treated with powdered by-products of silkworm larvae are displayed in Figure 2. The immunofluorescence results revealed the markedly increased expression of p65 along with the noticeable expression of p65 nuclear translocation in the TNF-α/INF-γ co-stimulated group compared to that in the control group (Figure 2A). However, no obvious difference was observed in the expression of p65 nuclear translocation between the HaCaT cells treated with SLPY at a 200 µg/mL concentration and the control. Moreover, all of the groups treated with silkworm by-products showed a decreased expression of p65 along with p65 nuclear translocation, suggesting that silkworm by-products may control the activation of the NF-kB signaling pathway in stimulated HaCaT cells. This effect was more pronounced at higher concentrations of silkworm by-products. Compared to other silkworm by-products, SLPY exhibited a promising effect in attenuating p65 nuclear translocation, especially at high concentrations. Therefore, high concentrations of SLPY effectively attenuated the p65 nuclear translocation in the HaCaT cells by controlling the activation of the NF-kB signaling pathway.

### 3.4. SLPY Downregulates Expression of Inflammation-Related Proteins

A Western blot analysis was performed to investigate the anti-inflammatory mechanisms of SLPY. The levels of p-p38 and p-ERK were measured in response to SLPY treatment using stimulated HaCaT cells. The control group exhibited significantly (*p* < 0.01) low p-p38 expression levels than the TNF-α/INF-γ-treated group. Figure 3 illustrates the marked suppression of p-p38 expression after SLPY treatment. Moreover, SLPY suppressed p-p38 expression in a concentration-dependent manner, where high concentrations exhibited a significant (*p* < 0.0001) reduction in expression. However, there were no significant differences between the expression level of the cells treated with 100 and 200 µg/mL SLPY. The p-ERK expression also followed a similar pattern: the TNF-α/INF-γ co-treated cells exhibited a significantly (*p* < 0.0001) higher expression compared to the control group. SLPY treatment decreased the elevated p-ERK expression levels, and only high concentrations of SLPY effectively decreased p-ERK expression. Moreover, the p-ERK expression in the cells treated with 100 and 200 µg/mL of SLPY was insignificant.

### 3.5. SLPY Ameliorates AD-like Skin Lesions in DNCB-Treated BALB/c Mice

The anti-AD effects of SLPY were assessed in an AD mouse model. AD-like skin lesions were induced by repeated DNCB application, and SLPY was topically applied along with the vehicle to assess the therapeutic efficacy of SLPY in AD. The control group, which was not subjected to repeated DNCB application, was free of the clinical symptoms of AD. Therefore, AD-like skin injuries were only observed in BALB/c mice after the repeated application of DNCB. In contrast, the AD group exhibited AD-like clinical features, including erythema, edema, dryness, and erosion. Moreover, plaques with excoriation and scaly patches were characteristic features of the AD group after three weeks of DNCB treatment (Figure 4A). However, the relief of the clinical symptoms of AD became a prominent feature in the SLPY-treated group. More importantly, high concentrations of SLPY did not cause any notable abnormalities in the dorsal skin. Similar results were observed with the topical application of Dermatop, where the clinical symptoms of AD markedly decreased after three weeks of treatment. Therefore, SLPY and Dermatop effectively ameliorated the AD-like skin injuries in the DNCB-treated mice. Figure 4B shows the average body weight of the mice in the corresponding treatment groups. As illustrated in the figure, there were no noticeable changes in the body weights of the mice in the different treatment groups.

The effect of AD on immune organs is well documented, as AD develops as a systemic immune response [22]. Thus, the weights of the spleen and lymph nodes were measured to assess the effect of SLPY treatment on the development of AD-like skin lesions. The AD group reported a significantly (*p* < 0.01) higher spleen weight than the control group. These findings are in accordance with the size of the spleen displayed in Figure 4A, where the AD group had a larger spleen size than the control. The size of the spleen gradually decreased with SLPY and Dermatop treatments. A similar pattern was observed for the size of the inguinal lymph nodes (Figure 4A). The size of the inguinal lymph nodes in the AD group was larger than that in the control group and gradually decreased with SLPY and Dermatop treatments. Dermatop application was more pronounced in reducing the size of the inguinal lymph nodes than the SLPY treatment. As shown in Figure 4C, the difference in the spleen weights between the AD and AD-V groups was insignificant. However, the spleen weights in the AD and treatment groups were significantly different based on SLPY and Dermatop treatment (*p* < 0.01). Therefore, SLPY and Dermatop are effective in attenuating the effects of atopic dermatitis. Figure 4D illustrates the changes in the axillary lymph nodes with different treatments. The weight of the axillary lymph nodes also followed a similar pattern to the spleen weight; the AD group exhibited a significantly higher spleen weight than the control group (*p* < 0.01). Moreover, high concentrations of SLPY and Dermatop effectively decreased the weight of the axillary lymph nodes. Contrarily, the low concentration of SLPY did not significantly differ from the AD group; only the high concentration of SLPY effectively decreased the weight of the axillary lymph nodes. Figure 4E shows the changes in the inguinal lymph nodes after treatment with SLPY and Dermatop. Dermatop treatment was more effective in decreasing the size of the inguinal lymph nodes than SLPY, and the inguinal lymph node size differed significantly with high concentrations of SLPY and Dermatop treatments.

### 3.6. Effect of SLPY on Inflammatory-Related Protein Expression in Skin Tissues of DNCB-Treated Mice

The effect of SLPY on inflammatory protein expression was assessed using Western blot analysis to confirm the mechanism of its anti-AD effect. Figure 5 shows the Western blot for DNCB-stimulated mice treated with SLPY. DNCB treatment markedly increased the protein expression level of TNF-α, iNOS, and Cox-2 compared with that in the control group. Moreover, SLPY treatment significantly decreased the protein expression level of TNF-α in a concentration-dependent manner. However, there was a significant (*p* < 0.001) difference between the TNF-α expression level of SLPY-H and the positive control: Dermatop. Similarly, SLPY treatment effectively decreased Cox-2 and iNOS expression, and SLPY-H significantly (*p* < 0.001) decreased the expression of iNOS compared to Dermatop. However, the Cox-2 expression levels in the SLPY and Dermatop-treated groups were insignificant.

### 3.7. SLPY Inhibits the Expression Levels of Inflammatory Cytokines in DNCB-Induced Mice

The ability of SLPY to alter the immune response was determined by assessing its effect on the expression of inflammatory cytokines. Figure 6 shows the effects of SLPY on the levels of inflammatory cytokines in the dorsal skin of DNCB-induced AD mice.

The effect of SLPY and Dermatop on TNF-α expression is displayed in Figure 6A. The level of TNF-α expression was significantly (*p* < 0.01) increased in the AD group compared to that in the control group. Moreover, the expression levels of TNF-α in the AD and ADV groups were insignificant. However, the SLPY and Dermatop treatment downregulated the expression of TNF-α. On the contrary, the TNF-α expression level of the low-concentration treatment of SLPY was insignificant. The level of TNF-α expression in the Dermatop and high SLPY concentration treatment was insignificant. As shown in Figure 6B, the SLPY treatment effectively downregulated IL-4 expression compared to that in the AD group. Similar to the TNF-α expression, the AD group significantly (*p* < 0.01) upregulated the IL-4 expression compared to the control. Dermatop was more effective in downregulating IL-4 expression than SLPY treatment. In contrast, the topical application of SLPY was more efficient in downregulating IL-10 expression than Dermatop (Figure 6D). Moreover, DNCB treatment upregulated the expression of IL-10 in the dorsal skin of the mice compared to that in the control. Figure 6C illustrates changes of IL-6 expression following SLPY treatment. As expected, the topical application of SLPY significantly (*p* < 0.05) downregulated IL-6 expression compared to the AD group. However, the topical application of Dermatop was more effective than SLPY in downregulating the expression level of IL-6. The IL-13 and IL-17 expression levels followed a similar pattern, and the topical application of SLPY and Dermatop effectively downregulated the expression of inflammatory cytokines. Moreover, the low concentration of SLPY effectively downregulated the expression of IL-17, and the difference between the low and high concentrations of SLPY and Dermatop treatment was insignificant (Figure 6F). Therefore, SLPY is a promising treatment for controlling the overexpression of inflammatory cytokines.

### 3.8. SLPY Reduces Pathological Damage in Skin Tissues of DNCB-Treated Mice

Figure 7 presents the mast cell infiltration and total and epidermal thicknesses of the dorsal skin of the DNCB-treated BALB/c mice compared to those of healthy mice. Hematoxylin and eosin staining was performed to assess the effect of SLPY treatment on the DNCB-treated skin lesions in AD mice. As illustrated in Figure 7A, the total and epidermal thicknesses markedly increased with DNCB treatment. There was a significant (*p* < 0.01) difference between the epidermal and total skin thicknesses between the AD and control groups. Moreover, there were no significant differences in the total and epidermal thickness between the AD and ADV groups. However, SLPY application noticeably decreased the total and epidermal skin thickness. High concentrations of SLPY were more effective than low concentrations in decreasing the total and epidermal thickness. As shown in Figure 7C, the epidermal thicknesses of the SLPY-H and Dermatop groups were insignificantly different. Similarly, the total skin thickness in the SLPY-H and Dermatop groups was not significantly different (*p* > 0.01) (Figure 7D). Therefore, the high concentrations of SLPY were more effective in reducing the pathological damage caused by atopic dermatitis.

The mast cell infiltration in the DNCB-induced inflamed skin, which is a characteristic feature of inflamed skin, was measured using toluidine blue staining. Therefore, we examined the effects of SLPY treatment on the mast cell infiltration of AD-like skin lesions in DNCB-treated mice. DNCB treatment significantly (*p* > 0.01) increased the mast cell count in the AD group compared to the control. However, SLPY treatment decreased the mast cell infiltration. Low concentrations of SLPY also significantly decreased the mast cell counts. However, high concentrations of SLPY were more effective at decreasing the mast cell infiltration. Furthermore, the mast cell counts in the SLPY-H and Dermatop groups were not significantly different (*p* > 0.01).

### 3.9. SLPY Application Reduced IgE and IgG2a Levels in DNCB-Treated AD Mice

Increased serum IgE levels are frequently observed in patients with AD. Therefore, in the present study, we assessed the serum IgE levels in the DNCB-stimulated AD mice. The results showed elevated levels of IgE in the DNCB-treated group compared to those in the control group. The topical application of SLPY markedly reduced the increase in serum IgE levels in the DNCB-treated mice (Figure 8A). The serum IgG2a levels followed a similar trend, with the DNCB-treated group displaying significantly higher levels of serum IgG2a than the control group. However, SLPY treatment gradually decreased the DNCB-induced increase in serum IgG2a levels. Furthermore, only the high-concentration SLPY treatment significantly (*p* < 0.05) decreased the serum IgG2a levels compared with the DNCB treatment group. Thus, the SLPY-H treatment was more effective than the positive control (Dermatop).

### 3.10. Amino Acid Composition of Powdered By-Products of Silkworm Larvae

The amino acid composition of powdered by-products of silkworm larvae from three different strains—Baekokjam (SLPB), Yeonnokjam (SLPY), and Golden Silk (SLPG)—was determined, and the results are shown in Figure 9. The amino acid compositions of the three strains were not markedly different (Figure 9); however, slight changes were observed in the quantity of each amino acid. Aspartic acid, threonine, serine, glutamic acid, glycine, alanine, methionine, tyrosine, leucine, histidine, and phenylalanine were the most abundant amino acids in SLPB, SLPY, and SLPG. In addition, these silkworm larval by-products also contained γ-aminobutyric acid (GABA). The silkworm by-product Yeonnokjam exhibited markedly higher levels of GABA than the Baekokjam and Golden Silk by-products. Moreover, high levels of isoleucine were evident in the silkworm by-product of the Yeonnokjam strain.

## 4. Discussion

Atopic dermatitis has gradually contributed to the global health burden; thus, several studies have focused on exploring natural sources to develop promising therapeutic agents to treat AD. Over the years, several studies have attempted to quantify the therapeutic efficacy of natural preparations using an AD-like mouse model with the typical clinical characteristics of AD [23]. Similarly, in this study, DNCB was used to successfully induce AD-like skin lesions in the dorsal skin of mice. DNCB treatment increased the epidermal thickness and inflammatory cell infiltration. These results are consistent with those of previous studies in which repeated DNCB application induced the clinical features of AD [21,24,25]. Skin thickening is one of the most prominent clinical features of AD. Thickened and hardened skin in AD is an outcome of inflammatory responses and continuous allergic conditions. SLPY application markedly decreased skin thickness, and thus exhibited inhibitory activity against hyperkeratosis. Similar findings have been reported in previous studies on *Lycopus lucidus* Turcz [26]. We employed this DNCB-induced mouse model to study the therapeutic efficiency of SLPY because silkworm pupae have numerous functional properties. Furthermore, studies have demonstrated the immune regulatory activity of proteins and hydrolysis peptides in silkworm pupae [27]. Therefore, SLPY has remarkable potential for the amelioration of AD. As expected, SLPY treatment attenuated the AD-like skin lesions in mice, as revealed by decreased levels of IgE and inflammatory cytokines and reduced epidermal thickness. Therefore, this study revealed the potential of the topical application of SLPY as a novel therapeutic drug treatment for AD.

Immune dysregulation and skin barrier dysfunction are well-known pathophysiological factors associated with AD. Among these, an imbalance in the Th1/Th2 cell ratio is a prominent feature of AD. Th1 cells largely produce IL-2 and IFN-γ, and IL-4 and IL-5 are mainly produced by Th2 cells. The Th1/Th2 imbalance causes skin inflammatory diseases, and in AD, this imbalance is skewed towards Th2. IL-17 secreted by Th17 cells also plays a pivotal role in AD [3]. Therefore, in this study, the effects of SLPY on the expression of inflammatory cytokines were explored to evaluate its therapeutic potential against AD. In vitro experiments demonstrated the potential of SLPY in downregulating the expression levels of TNF-α, IL-1β, IL-8, and Cox-2 in TNF-α/IFN-γ-treated HaCaT cells. Additionally, SLPY treatment downregulated the expression of IL-4, IL-6, IL-10, IL-13, and IL-17.

To further elucidate the underlying mechanism of SLPY in ameliorating AD-like skin lesions, the nuclear translocation of NF-κB was studied in vitro. Most cells contained the protein complex NF-κB, which mediates diverse processes, including inflammation, immune response, cell growth, and apoptosis. The activation of NF-κB is characterized by the degradation of the inhibitory proteins IκBα and IκBβ. Moreover, the nuclear translocation of p65, p50, cRel, RelB, and p52 subunits also featured in NF-κB activation. In particular, the nuclear translocation of NF-κB transcriptomic subunit p65 initiates the activation of pro-inflammatory genes, including TNF-α and IL1β [28]. The current study revealed increased levels of NF-κB p65 nuclear translocation in HaCaT cells treated with TNF-α/INF-γ. TNF-α/INF-γ treatment synergistically activates the NF-κB pathway, leading to the production of pro-inflammatory cytokines. This was evidenced by the PCR results, where the overexpression of TNF-α and IL1β occurred with the co-treatment of TNF-α/INF-γ. Treatment with the silkworm by-product gradually decreased the nuclear translocation of NF-κB p65, and SLPY was effective in attenuating the nuclear translocation of NF-κB p65. The silkworm by-product treatment gradually decreased the overexpression of the pro-inflammatory cytokines TNF-α and IL1β. Similar results were reported in a previous study where *Soshiho-tang*, a well-documented traditional herbal medicine, decreased NF-κB p65 translocation in HaCaT cells [29].

Elevated levels of IgE and infiltration of inflammatory cells, mainly mast cells, are notable features of AD. Moreover, the infiltration of mast cells has been identified as a prominent clinical characteristic of patients with AD. Mast cell activation is strongly associated with allergic inflammation, leading to the expression of inflammatory mediators. Studies have shown that an increased number of mast cells is a common feature of human AD and DNCB-induced AD in mice [26]. Therefore, in the current study, mast cell infiltration was examined by H&E staining. The results revealed elevated levels of mast cell infiltration in the AD group and a marked reduction in mast cell infiltration in the SLPY-treated group. Moreover, the results of this study further confirmed the reduced levels of IgE in the serum of the SLPY-treated mice. In the acute stage of AD, Th-2 cell-related allergic inflammation becomes a prominent feature, leading to the increased production of IL-6 and IgE. The activation of mast cells initiates the binding of IgE to the IgE binder FcεRI [26]. The AD group exhibited elevated levels of IL-6, whereas SLPY treatment gradually decreased these levels. Similarly, the elevated levels of IgE in the DNCB-induced AD group were significantly decreased by SLPY treatment.

Currently, the activation of NF-κB is defined as a critical factor in inflammatory skin disease; therefore, the inhibition of NF-κB expression has become of utmost importance in inflammatory disease-related therapies. The activation of NF-κB is strongly linked with the mitogen-activated protein kinase (MAPK) signaling pathway [24]. Therefore, in the current research, the effect of SLPY treatment on MAPK-related protein expression in TNF-α/INF-γ co-stimulated HaCaT cells was determined. The SLPY treatment effectively reduced the levels of p-p-38 and p-ERK compared to those in TNF-α/INF-γ-treated cells. Thus, SLPY can effectively be utilized as a potential therapeutic drug in the treatment of AD by inactivating the MAPK pathway and subsequently inhibiting NF-κB expression.

The amino acid composition of the silkworm by-products was determined using an amino acid analyzer to determine the most prominent amino acids present in each by-product. SLPY exhibited high levels of GABA, which plays a major role in modulating the immune response by downregulating the expression of pro-inflammatory cytokines. Moreover, GABA controls immune cell proliferation. Therefore, the anti-AD effects of SLPY may be attributed to the presence of GABA [30]. Moreover, SLPY exhibits high histidine levels. Studies have proven the effectiveness of L-histidine supplementation in controlling AD. L-histidine supplementation is a promising method for increasing the skin barrier protein filaggrin [31]. Furthermore, markedly higher levels of isoleucine were also reported in SLPY than in SLPB and SLPG. The anti-inflammatory activity of isoleucine has been reported in previous studies and thus exhibits potential utilization in AD management [32].

## 5. Conclusions

In this study, we investigated the anti-AD effect of a powdered by-product of silkworm larvae from the Yeonnokjam strain and demonstrated the potential use of SLPY as a sustainable natural source for the treatment of AD. The results of this study demonstrated that SLPY treatment decreases the overexpression of inflammatory cytokines and proteins. Moreover, the increased levels of serum IgE and IgG2a in DNCB-treated mice were effectively decreased with the topical application of SLPY. Moreover, SLPY treatment markedly decreased the clinical features of AD, including epidermal thickness. Furthermore, SLPY treatment inhibited the nuclear translocation of NF-κB p65. As confirmed by the results of the current study, the powdered by-product of silkworm larvae from the Yeonnokjam strain exhibits considerable potential as a therapeutic agent for the treatment of AD. However, further studies are crucial to identify the active compounds present in SLPY and their effects on AD.

## Figures and Tables

**Figure 1 nutrients-15-01775-f001:**
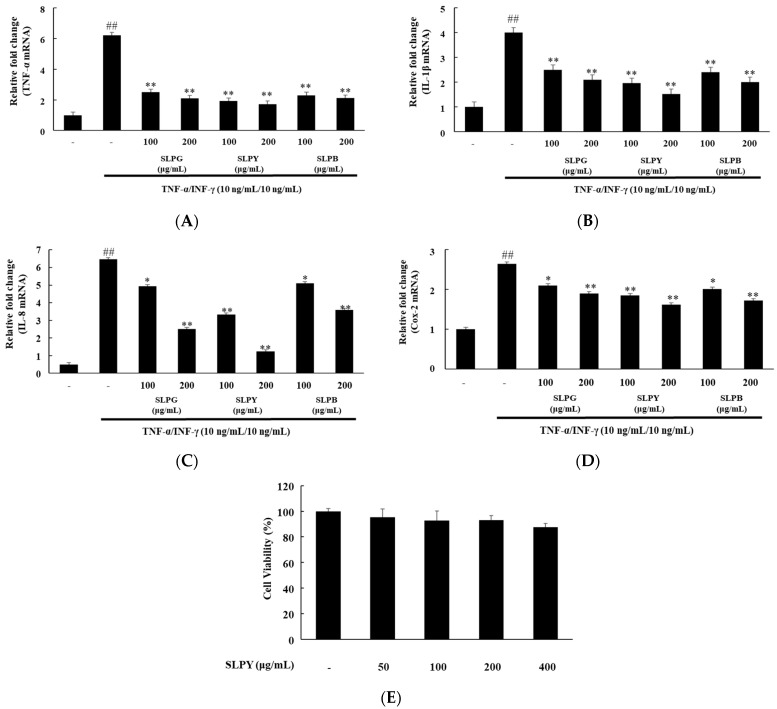
Effect of different silkworm by-products on levels of inflammatory cytokine expression in HaCaT cells. (**A**) TNF-α, (**B**) IL-1β, (**C**) IL-8, and (**D**) Cox-2 mRNA expression in HaCaT cells treated with SLPG, SLPY, and SLPB at 100 and 200 μg/mL concentrations, followed by TNF-α (10 ng/mL)/INF-γ (10 ng/mL) co-treatment for 6 h. (**E**) Percentage of HaCaT cell viability measured using MTT assay. Following the treatment with SLPY for 24 h, the viability of HaCaT cells was compared with that of the control (100%). Values represent the mean ± SD of three replicates of viable cell percentage compared to the control. ## *p* < 0.01 vs. the corresponding control group and ** *p* < 0.01, * *p* < 0.05 vs. the corresponding TNF-α/INF-γ treated group. TNF: tumor necrosis factor; IL: interleukin; Cox-2: cyclooxygenase-2; SLPG: steamed and freeze-dried powdered by-product of mature silkworm larvae of the Golden Silk strain; SLPY: steamed and freeze-dried powdered by-product of mature silkworm larvae of the Yeonnokjam strain; SLPB: steamed and freeze-dried powdered by-product of mature silkworm larvae of the Baekokjam strain.

**Figure 2 nutrients-15-01775-f002:**
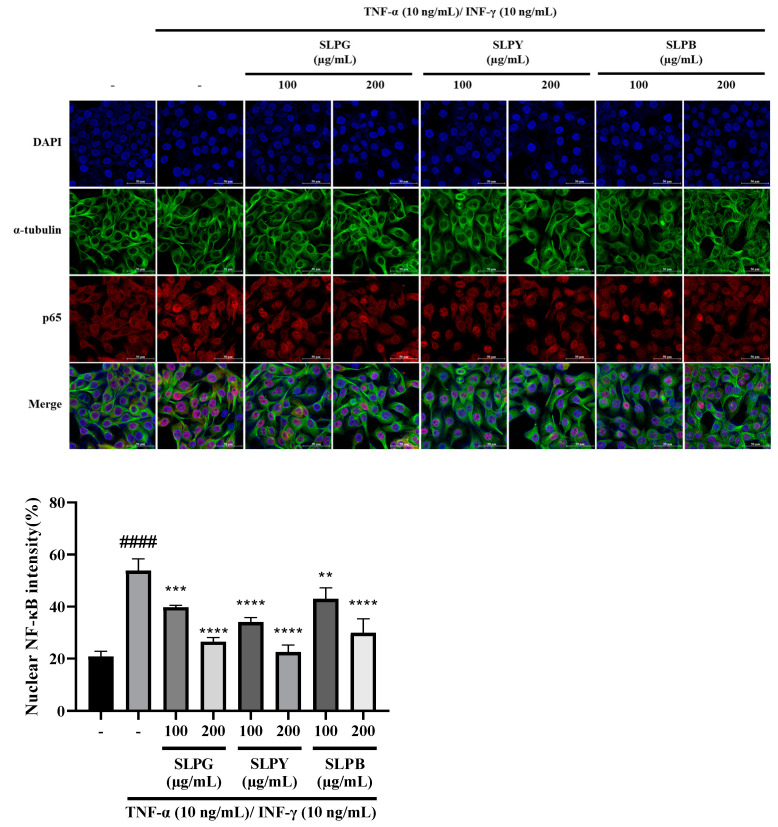
Immunofluorescence staining of TNF-α/INF-γ co-stimulated HaCaT cells treated with powdered by-products of silkworm larvae. NF-κB activation was assessed by immunostaining for p65 intracellular localization (red) overlaid with 4′,6-diamidino-2-phenylindole dihydrochloride (DAPI) nuclear stain (blue); the green color represents α-tubulin. SLPG: steamed and freeze-dried powdered by-product of mature silkworm larvae of the Golden Silk strain; SLPY: steamed and freeze-dried powdered by-product of mature silkworm larvae of the Yeonnokjam strain; SLPB: steamed and freeze-dried powdered by-product of mature silkworm larvae of the Baekokjam strain. #### *p* < 0.0001 vs. the corresponding control group and ** *p* < 0.01, *** *p* < 0.001, **** *p* < 0.0001 vs. the corresponding TNF-α/INF-γ treated group.

**Figure 3 nutrients-15-01775-f003:**
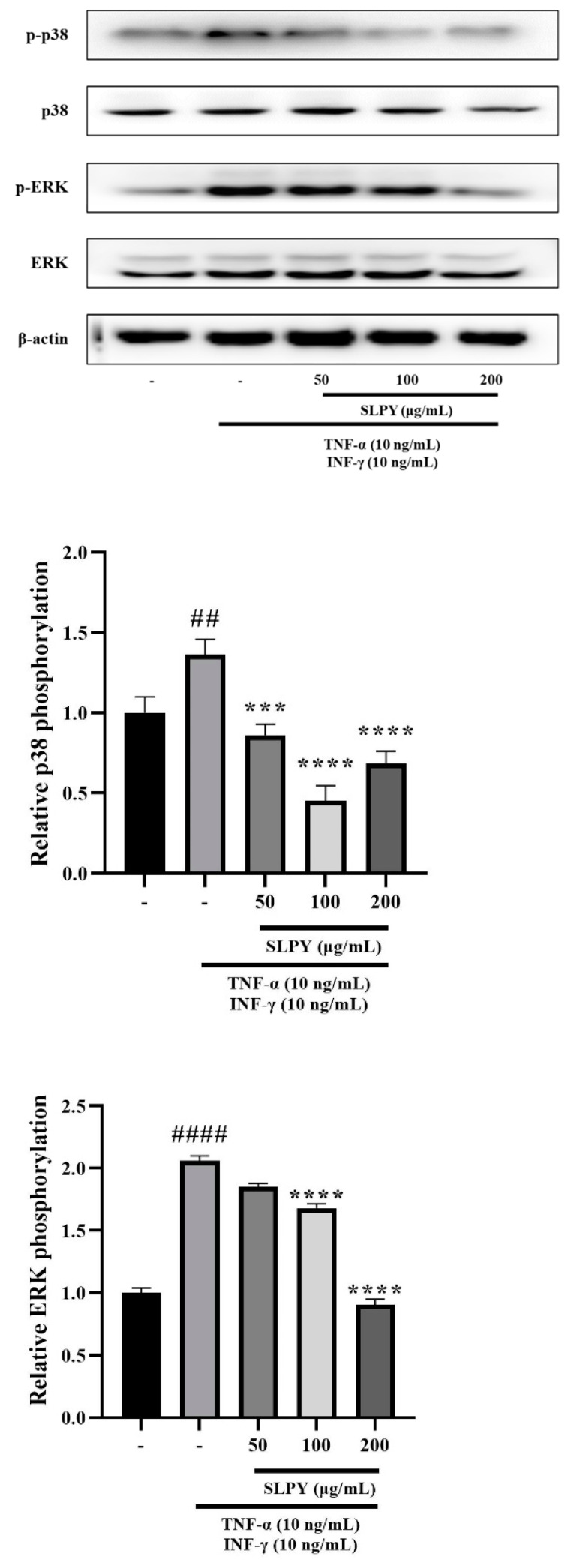
Effect of SLPY on expression levels of proteins linked to the mitogen-activated protein kinase (MAPK) signaling pathway in HaCaT cells. Cells were tested for three concentration levels of SLPY: 50, 100, and 200 µg/mL after stimulation with interferon (IFN)-γ and tumor necrosis factor (TNF)-α at 10 ng/mL for 6 h. Values represent the mean ± SD of at least three replicates. ## *p* < 0.01 and #### *p* < 0.0001 vs. the corresponding control group and *** *p* < 0.001, **** *p* < 0.0001 vs. the corresponding TNF-α/INF-γ group. SLPY: steamed and freeze-dried powdered by-product of mature silkworm larvae of the Yeonnokjam strain.

**Figure 4 nutrients-15-01775-f004:**
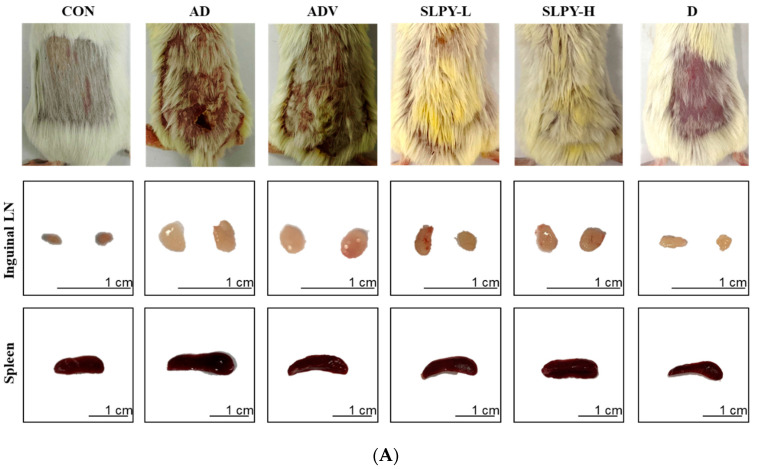
Effect of SLPY topical application on AD-like skin injuries in DNCB-induced BALB/c mice. (**A**) Clinical characteristics, size of inguinal lymph node, and size of spleen of DNCB-treated AD mice. (**B**) Body weight, (**C**) spleen weight, (**D**) axillary lymph node weight, and (**E**) inguinal lymph node weight of control, AD, ADV, SLPY, and Dermatop-treated groups. The values are expressed as mean ± SD of at least three replicates. ## *p* < 0.01 vs. the corresponding control group and ** *p* < 0.01, * *p* < 0.05 vs. the corresponding treatments groups. CON: non-treatment group; AD: 2,4-dinitrochlorobenzene (DNCB)-induced group; ADV: combined treatment of 1% DNCB and vehicle; SLPY-L: combined treatment of 1% DNCB and 50 mg/Kg SLPY; SLPY-H: combined treatment of 1% DNCB and 100 mg/Kg SLPY; D: combined treatment of 1% DNCB and Dermatop (0.25% (*w/w*).

**Figure 5 nutrients-15-01775-f005:**
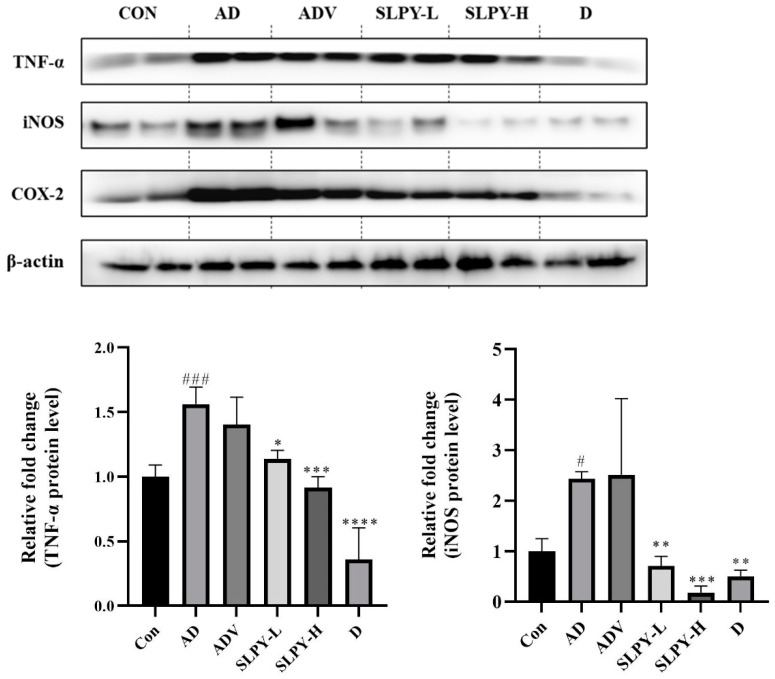
Effect of SLPY on level of inflammatory protein expression in skin tissues of DNCB-treated mice. Mice were challenged with DNCB to induce AD-like skin lesions and tested with SLPY to assess the changes in the levels of inflammation-related protein expression: tumor necrosis factor-α (TNF-α), cyclooxygenase (Cox-2), and inducible nitric oxide synthase (iNOS). Values represent the mean ± SD of three replicates. #### *p* < 0.0001, ### *p* < 0.001, and # *p* < 0.05 vs. the corresponding control group and **** *p* < 0.0001, *** *p* < 0.001, ** *p* < 0.01, and * *p* < 0.05, vs. the corresponding AD group. SLPY: steamed and freeze-dried powdered by-product of mature silkworm larva of the Yeonnokjam strain. CON: non-treatment group; AD: 2,4-dinitrochlorobenzene (DNCB)-induced group; ADV: combined treatment of 1% DNCB and vehicle; SLPY-L: combined treatment of 1% DNCB and 50 mg/Kg SLPY; SLPY-H: combined treatment of 1% DNCB and 100 mg/Kg SLPY; D: combined treatment of 1% DNCB and Dermatop (0.25% (*w/w*).

**Figure 6 nutrients-15-01775-f006:**
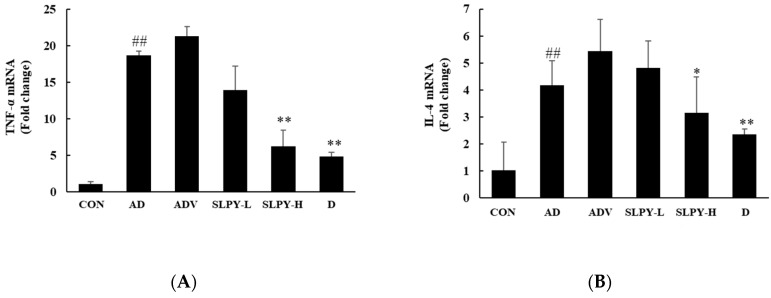
Effect of SLPY on the levels of inflammatory cytokine expression in the dorsal skin of DNCB-treated AD mice. (**A**) TNF-α, (**B**) IL-4, (**C**) IL-6, (**D**) IL-10, (**E**) IL-13, and (**F**) IL-17 mRNA levels of DNCB-induced AD mice with the treatment of SLPY. Data displayed as mean ± SD of at least three replicates. ## *p* < 0.01 vs. the corresponding control group and ** *p* < 0.01, * *p* < 0.05 vs. the corresponding AD group. CON: non-treatment group; AD: 2,4-dinitrochlorobenzene (DNCB)-induced group; ADV: combined treatment of 1% DNCB and vehicle; SLPY-L: combined treatment of 1% DNCB and 50 mg/Kg SLPY; SLPY-H: combined treatment of 1% DNCB and 100 mg/Kg SLPY; D: combined treatment of 1% DNCB and Dermatop (0.25% (*w/w*).

**Figure 7 nutrients-15-01775-f007:**
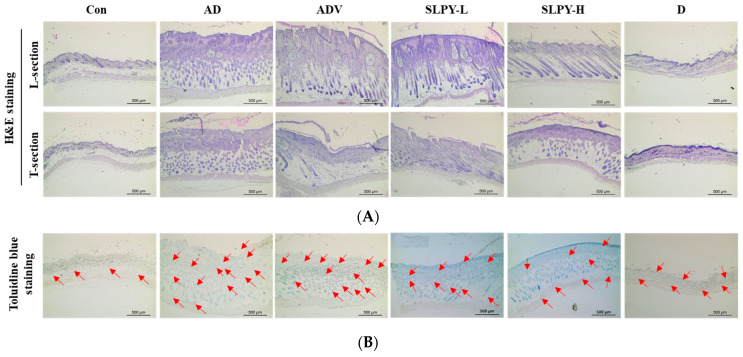
Effect of topical application of SLPY by-product on the histological features of dorsal skin of DNCB-treated AD mice. (**A**) Representative photographs of hematoxylin and eosin (H&E)-stained skin tissues isolated from SLPY-treated AD mice at ×50 magnification (scale bar: 500 µm). (**B**) Representative photographs of toluidine blue-stained dorsal skin tissues of SLPY-treated AD mice at ×50 magnification (scale bar: 500 µm). (**C**) Epidermal thickness of DNCB-treated BALB/c mice tested with SLPY against control and Dermatop. (**D**) Total skin thickness of DNCB-treated BALB/c mice tested with SLPY against the control and Dermatop. (**E**) Mast cell number of DNCB-treated BALB/c mice tested with SLPY against the control and Dermatop. Red arrows pinpoint the mast cells present in the toluidine blue-stained dorsal skin tissues. Data represented as mean ± SD of at least three replicates. Significant differences in each treatment group indicated based on Dunnett’s multiple comparison post hoc analysis. ## *p* < 0.01 vs. the corresponding control group and ** *p* < 0.01, * *p* < 0.05 vs. the corresponding AD group. CON: non-treatment group; AD: 2,4-dinitrochlorobenzene (DNCB)-induced group; ADV: combined treatment of 1% DNCB and vehicle; SLPY-L: combined treatment of 1% DNCB and 50 mg/Kg SLPY; SLPY-H: combined treatment of 1% DNCB and 100 mg/Kg SLPY; D: combined treatment of 1% DNCB and Dermatop (0.25% *w/w*).

**Figure 8 nutrients-15-01775-f008:**
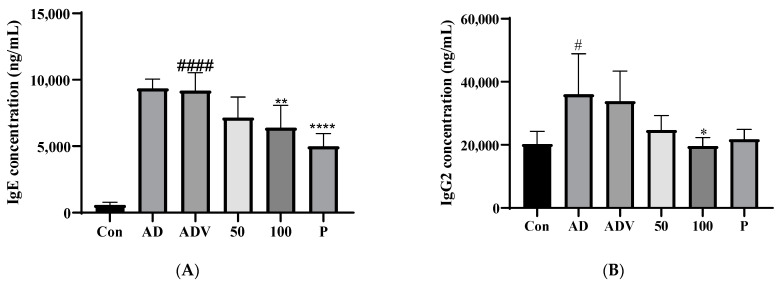
Effect of SLPY on IgE and IgG2 concentration of DNCB-treated AD mice. (**A**) Serum IgE level of AD mice against the control and SLPY treatment groups. (**B**) Serum IgG2a level of AD mice against the control and SLPY treatment groups. Data represent mean ± SD of at least three replicates. Significant differences in each treatment group indicated based on Dunnett’s multiple comparison post hoc analysis. #### *p* < 0.0001 and # *p* < 0.05 vs. the corresponding control group and **** *p* < 0.0001, ** *p* < 0.01 and * *p* < 0.05, vs. the corresponding AD group. CON: non-treatment group; AD: 2,4-dinitrochlorobenzene (DNCB)-treated group; ADV: combined treatment of 1% DNCB and vehicle; SLPY-L: combined treatment of 1% DNCB and 50 mg/Kg SLPY; SLPY-H: combined treatment of 1% DNCB and 100 mg/Kg SLPY; D: combined treatment of 1% DNCB and Dermatop (0.25% *w/w*).

**Figure 9 nutrients-15-01775-f009:**
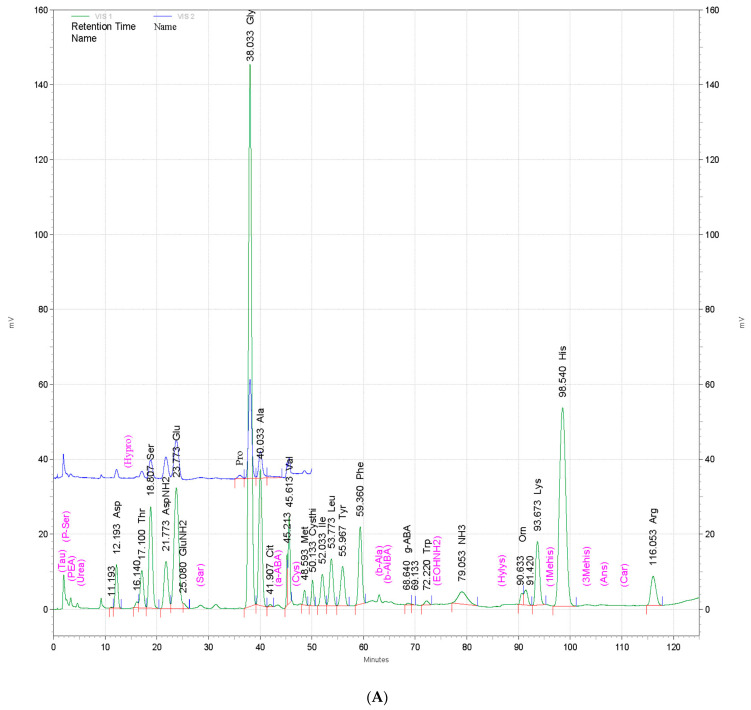
Amino acid composition of powdered by-products of mature silkworm larvae of (**A**) Baekokjam (SLPB), (**B**) Yeonnokjam (SLPY), and (**C**) Golden Silk (SLPG) strains. Color coded by the machine according to wavelength of the UV detector. Green color represents VIS1 (570 nm) and blue color represents VIS2 (440 nm).

**Table 1 nutrients-15-01775-t001:** Sequence of the primers used in RT-qPCR.

Gene	Primer	Sequence
GAPDH	Forward	5’-CCCCTGGCCAAGGTCATCCATGACAACTTT-3’
	Reverse	5’-GGCCATGAGGTCCACCACCCTGTTGCTGTA-3’
TNF-α	Forward	5’-CCCTCCAGTTCTAGTTCTATC-3’
	Reverse	5’-GGGGAAAGAATCATTCAACCAG-3’
IL-1β	Forward	5’-ACGATGCACCTGTACGATCA-3’
	Reverse	5’-TCTTTCAACACGCAGGACAG-3’
IL-8	Forward	5’-ACAGCAGAGCACACAAGCTT-3’
	Reverse	5’-CTGGCAACCCTACAACAGAC-3’
Cox-2	Forward	5’-AACAGGAGCATCCTGAATGG-3’
	Reverse	5’-GGTCAATGGAAGCCTGTGAT-3’

## Data Availability

All data are presented in the manuscript.

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
