# Peer review of "Use of a Silkworm (Bombyx mori) Larvae By-Product for the Treatment of Atopic Dermatitis: Inhibition of NF-κB Nuclear Translocation and MAPK Signaling"

_nutrients, 2023, doi:10.3390/nu15071775_

Round 1

Reviewer 1 Report

The work is well structured, and clear. However, some minor revisions are rquired:

Confocal microscope magnification used is required in methods and resultsScale bar in confocal microscopy images must be added

There is no correspondance between the magnification used for histological analysis in methods (100x) and results (50x); is there any reason?

For paragraph 3.2, can you show the increased expression of p65 microscopically observed (line 328-339) as hystograms of fluorescence increased intensity?

Review the English in the whole manuscript for some typos, es line 700-703

Author Response

Dear Reviewer,

We gratefully thank you for the valuable comments given to improve the quality of the manuscript. 

Reviewer 2 Report

In the manuscript “Anti-Atopic Dermatitis Effect of Silkworm (Bombyx Mori) Lar-2 vae By-Product: Inhibition of NF-κB Nuclear Translocation and 3 MAPK Signaling”, Fan et al investigated the potential effect of by-product of silkworm larval powder on the treatment of AD in vitro and in vivo.  The author claimed that SLPY could induced inhibits the expression of inflammatory cytokines in both transcriptional and translational levels and easing the clinical symptoms of AD in vivo. Furthermore, the author found that SLPY could inhibited the nuclear translocation of NF-κb p65. The topic is interesting, but the author's carelessness is reflected in every nook and cranny of the manuscript. There are several comments to help the authors improve their manuscript.

Major comments:

1. The language should be improved significantly.

2. The composition of Silkworm (Bombyx mori) larvae by-product was complicated, including the amino acid detected in this manuscript. What are the active ingredients? A comprehensive summarize or explore is required.

3. For all I know, the silkworm protein could induce allergic reactions in some of individuals. Did the author observe the similarity symptoms or side effect on the animal model?

4. The full name of the abbreviate should be provided when it first occurs, even in the abstract section.  For example, DNCB.

5. Figure 1 and table 2 (labeled by the author) Experimental design for DNCB-induced BALB/c mice & Primer sequence of the respective gene used in qRT-PCR analysis.Those figure and table should be moved to supplementary data.

6. There should be two band of the p-ERK.

Minor comments:

Some of the minor deficiencies are listed below although this is by no means an exhaustive list.

1. [10], [11]; [12], [11], [10], [13] …….

2. Line 130, 1×105, superscript

3. Line 130, CO2, subscript

4. Line 147, 2– (ΔCt of target gene−ΔCt of GAPDH gene),

5. Line 153, 5×103, superscript

6. “Cell Viability by (3-(4,5-dimethylthiazol-2-yl)-2,5-diphenyltetrazolium bromide) MTT Assay” should be “Cell Viability by 3-(4,5-dimethylthiazol-2-yl)-2,5-diphenyltetrazolium bromide (MTT) Assay.”

7. Leave a space after the 1:300. For example, line 168, 1:300; Line 147, 1:1000.

8. Insert spaces in a big number to make it more readable. For example, line 147, 1:1000, should be change into “1 000”

Author Response

Dear Reviewer,

We greatly thank you for the valuable comments you have made to improve the quality of the manuscript.

Round 2

Reviewer 2 Report

Thank you for the author's reply. I am pleased to note that all of my comments have been addressed and the manuscript has been significantly improved.